# Whole-Genome Differentially Hydroxymethylated DNA Regions among Twins Discordant for Cardiovascular Death

**DOI:** 10.3390/genes12081183

**Published:** 2021-07-29

**Authors:** Jun Dai, Ming Leung, Weihua Guan, Han-Tian Guo, Ruth E. Krasnow, Thomas J. Wang, Wael El-Rifai, Zhongming Zhao, Terry Reed

**Affiliations:** 1Department of Public Health, College of Health Sciences, Des Moines University, Des Moines, IA 50312, USA; 2Institute for Personalized Medicine, Penn State College of Medicine, Hershey, PA 17033, USA; mleung@pennstatehealth.psu.edu; 3Division of Biostatistics, University of Minnesota School of Public Health, Minneapolis, MN 55455, USA; wguan@umn.edu; 4Bioinformatics and Computational Biology Undergraduate Program, Iowa State University, Ames, IA 50011, USA; hantianguo94@gmail.com; 5Center for Health Sciences, SRI International, Menlo Park, CA 94025, USA; ruth.krasnow@gmail.com; 6Department of Internal Medicine, University of Texas Southwestern Medical Center, Dallas, TX 75390, USA; Thomas.Wang@UTSouthwestern.edu; 7Department of Surgery, University of Miami Miller School of Medicine, Miami, FL 33136, USA; welrifai@med.miami.edu; 8Center for Precision Health, School of Biomedical Informatics, The University of Texas Health Science Center at Houston, Houston, TX 77030, USA; Zhongming.Zhao@uth.tmc.edu; 9Human Genetics Center, School of Public Health, The University of Texas Health Science Center at Houston, Houston, TX 77030, USA; 10Department of Medical and Molecular Genetics, Indiana University School of Medicine, Indianapolis, IN 46202, USA; treed@iupui.edu

**Keywords:** hydroxymethylation, twins, monozygotic, dizygotic, cardiovascular disease

## Abstract

Epigenetics is a mechanism underlying cardiovascular disease. It is unknown whether DNA hydroxymethylation is prospectively associated with the risk for cardiovascular death independent of germline and common environment. Male twin pairs middle-aged in 1969–1973 and discordant for cardiovascular death through December 31, 2014, were included. Hydroxymethylation was quantified in buffy coat DNA collected in 1986–1987. The 1893 differentially hydroxymethylated regions (DhMRs) were identified after controlling for blood leukocyte subtypes and age among 12 monozygotic (MZ) pairs (Benjamini–Hochberg False Discovery Rate < 0.01), of which the 102 DhMRs were confirmed with directionally consistent log_2_-fold changes and *p* < 0.01 among additional 7 MZ pairs. These signature 102 DhMRs, independent of the germline, were located on all chromosomes except for chromosome 21 and the Y chromosome, mainly within/overlapped with intergenic regions and introns, and predominantly hyper-hydroxymethylated. A binary linear classifier predicting cardiovascular death among 19 dizygotic pairs was identified and equivalent to that generated from MZ via the 2D transformation. Computational bioinformatics discovered pathways, phenotypes, and DNA motifs for these DhMRs or their subtypes, suggesting that hydroxymethylation was a pathophysiological mechanism underlying cardiovascular death that might be influenced by genetic factors and warranted further investigations of mechanisms of these signature regions in vivo and in vitro.

## 1. Introduction

Gene expression plays a role in the development of cardiovascular disease [1]. Without changing genetic codes, DNA hydroxymethylation is one of the epigenetic mechanisms that regulate gene expression through the activation of genes and executes functions independent of methylation [1,2]. It is known that genes can also affect epigenetic modifications [3].

5-hydroxymethylcytosine (5hmC), the “sixth” base in mammalian genomic DNA [4], results from the demethylation of 5-methylcytosine (5mC) through Fe(II) α-ketoglutarate-dependent hydroxylation catalyzed by the ten-eleven translocation (TET) family proteins [5]. An animal experiment demonstrated genome-wide hydroxymethylation, particularly in intronic regions, was one epigenetic mechanism underlying the development of dilated cardiomyopathy [6]. Previous human studies of hydroxymethylation and cardiovascular hard outcomes were few. A prior case-control study of myocardial infarction in the Chinese elderly aged 70 to 88 years demonstrated a modest positive correlation between global 5hmC levels in peripheral mononuclear cells and coronary atherosclerosis measured with the Gensini severity score [7]. Another case-control study of stable coronary heart disease and acute myocardial infarction showed the diagnostic and predictive property of 5hmC signature in circulating cell-free DNA for coronary heart disease [8]. However, both case-control studies were retrospective and unable to control for potential genetic confounding. Additionally, it is unknown if circulating whole-genome 5hmC was prospectively associated with long-term cardiovascular death independent of genetic and shared environmental influences and blood leukocyte composition in humans. The paucity of this fundamental knowledge impedes the understanding of hydroxymethylation as an epigenetic mechanism underlying the development of cardiovascular disease.

A nested case-co-twin-control design is a specific type of the 1:1 individually matched, nested case-control design. This design uses twins discordant for an outcome in a prospective twin cohort to dissect the prospective association between exposure and outcomes from genes and environment shared between co-twins of a twin pair [9]. The term “germline” [10] describes genetic factors shared between co-twins that are inherited from parents, including genome sequence and epigenetic modifications. Co-twins share germline and numerous environmental factors; thus, they are naturally matched for these factors [9]. Environmental factors common to co-twins, i.e., common or shared environment, can exist throughout their whole life course or in certain periods of their life. Common environmental factors can be both within and outside of the family [9]. Common familial factors include environmental factors that are shared between co-twins as they grow up and can have a life-long influence, such as the family history of coronary heart disease [9].

Using twins discordant for cardiovascular death included from the prospective 45-year National Heart, Lung, and Blood Institute (NHLBI) Twin Study, we aimed to characterize whole-genome differentially hydroxymethylated regions (DhMRs) in relation to the extended long-term risk of cardiovascular death as the outcome, independent of influences of germline and common environment.

## 2. Materials and Methods

### 2.1. Study Population

As has been shown before [9,11,12], the prospective NHLBI Twin Study was initiated in 1969 and enrolled 514 middle-aged, white, male, veteran twin pairs (1028 men, 254 MZ and 260 DZ twin pairs [13]) from the National Academy of Sciences-National Research Council Veteran Twin Registry, who lived within 200 miles of five research centers: Framingham, Massachusetts; San Francisco, California; Davis, California; Los Angeles, California; and Indianapolis, Indiana. The twins were born between 1917 and 1927 and were 43–56 years of age at baseline examination (i.e., exam 1, 1969–1973). All twins were physically examined at baseline and during follow-up with the well-established Framingham Heart Study protocol to ensure the uniform examination of all the twins by experienced cardiovascular epidemiologists. Zygosity was ascertained by eight red blood cell antigen groups (serotyping 22 erythrocyte antigens) in the 1960s and a variable number of tandem repeat DNA markers in the 1980s [13]. The NHLBI Twin Study was approved by the Institutional Review Board at each examination site, and all twins provided written informed consent.

### 2.2. Study Samples

In this study, we used a prospective nested case-co-twin-control design that included twin pairs discordant for cardiovascular death. This design was a specific type of the 1:1 individually matched, nested case-control design, in which one co-twin within a twin pair was the case, and his co-twin brother was the control. In the nested case-control design [14], control has not developed the disease by the time of disease occurrence in the case (index date) and may later become a case [14]. Therefore, in a twin pair discordant for the outcome (i.e., cardiovascular death), a co-twin who had the outcome is the case twin, while the co-twin who did not have the outcome on the event date of the case twin (index date) [15] is the control twin. We used the following definitions to identify discordant twin pairs. The primary definition of a twin pair discordant for total cardiovascular death was one where a co-twin died from cardiovascular disease (CVD) and his co-twin brother who did not die from it or died from it at least one year later. Thus, the co-twin who died first would be considered the case, and his brother would be the control. A more stringent definition was one where a co-twin died from the disease (case), and his co-twin brother did not die from it by the end of the follow-up (control). Inclusion criteria included: (1) available buffy coat DNA ≥ 400 microgram on the DNA inventory of Veteran Twin Samples after exam 5 (1999–2001) for each co-twin of a twin pair, and (2) available data on vital status, dates of death, and causes of death through December 31, 2010. We performed stratified random sampling for twins discordant for coronary heart death and non-coronary heart cardiovascular death by zygosity, separately. Based on the updated vital data in 2014, we included 19 monozygotic (CVD-dMZ) and 19 dizygotic (CVD-dDZ) twin pairs discordant for cardiovascular death from the NHLBI Twin Study by the end of 2014 in this study. This report study was approved by the Institutional Review Boards of Vanderbilt University and Des Moines University.

### 2.3. DNA Sample Collection

Buffy coat DNA samples were collected at exam 3 (1986–1987). Whole blood was drawn from the antecubital vein into EDTA tubes after an overnight fast and immediately placed on ice. Buffy coats were obtained and used to extract DNA. Spectral analysis was used to determine the quantity and quality of the extracted DNA. DNA samples were stored at −70 °C. All samples were labeled by the study number only.

### 2.4. Genome-Wide Methylation (5mC) Measures

Genome-wide methylation was measured with the Illumina Infinium^®^ HumanMethylation450 (450 K) BeadChip (Illumina, Inc. San Diego, CA, USA) following the established Illumina protocol. Co-twin samples were processed in the same analytical run without known the disease status to minimize measurement error. All samples passed quality control (missing percentage < 1.5%) [16]. CpGs with greater than 10% missing data were removed (452 CpGs removed) [16].

### 2.5. Measurement of Whole-Genome Hydroxymethylation (5hmC)

Whole-genome 5hmC profiling was performed using a published enrichment-based hMe-Seal-seq method with genome-wide coverage to detect 5hmC signals [17]. The published works have demonstrated that there is no bias with hMe-Seal and this method is more sensitive than the antibody-based immunoprecipitation approach [17,18,19]. It has been widely used to profile 5hmC in different tissues and cell types [18,20]. Co-twin samples were processed in the same analytical run without known the disease status to minimize measurement error.

#### 2.5.1. Genomic DNA Preparation

Genomic DNA was isolated in digestion buffer (100 mM Tris-HCl, pH 8.5, 5 mM EDTA, 0.2% SDS, 200 mM NaCl), Proteinase K (Thermo Fisher Scientific Baltics location, Vilnius, Lithuania, cat# EO0491) treatment at 55 °C for overnight. On the second day, phenol:chloroform:isoamyl alcohol (25:24:1 saturated with 10 mM Tris, pH 8.0, 1 mM EDTA) (Sigma-Aldrich, cat# P-3803) was added to samples, mixed completely, and centrifuged for 10 min at 12,000 rpm. The aqueous layer solution was transferred into a new Eppendorf tube and precipitated with 600 mL isopropanol. The pellet was washed with 75% ethanol, air-dried, and eluted with nuclease-free water (Ambion).

#### 2.5.2. 5hmC Capture and Sequencing

We employed a previously established chemical labeling and affinity purification method coupled with high-throughput sequencing (hMe-Seal) to profile the genome-wide distribution of 5hmC [17]. Briefly, genomic DNA was sonicated using a Covaris Ultrasonicator, yielding an average size of sonicated DNA fragments of ∼200 bp. Two micrograms of fragmented DNA were used for the hMe-Seal assay. For the hMe-Seal assay, the T4 bacteriophage β-glucosyltransferase was used to transfer a chemically modified glucose, 6-N3-glucose, onto the hydroxyl group of 5hmC. After attaching a biotin tag to the 6-N3-glucose, the azide group was chemically modified for biotin detection, then 5hmC-containing DNA fragments were high-affinity captured for deep sequencing.

The captured DNA fragments containing 5hmC were used for library preparation using NEBNext® ChIP-Seq Library Prep Master Mix Set for Illumina (New England Biolabs® Inc., Ipswich, Massachusetts, U.S., Cat# E6240L) per the manufacturer’s instructions with some modifications, i.e., no size selection and a 2.5-fold decrease in the quantity of the final PCR primers (Universal primer and Index primer) for PCR-based final enrichment of the library.

Single-end sequencing was performed in a depth of 25 million reads. Image processing and sequence extraction were performed using the standard Illumina pipeline. The single best alignment of raw sequence reads that passed the filtering to the reference genome (hg19) was performed using the Bowtie2 (version 2.3.1) [21].

### 2.6. Assessment of Covariates

Through in-person interviews and physical examinations, data on all other cardiovascular risk factors were collected [12,13,22]. Data on age, years of education, marital status, and smoking status were collected. Weight and height were measured. Systolic and diastolic blood pressures were measured using a standard mercury sphygmomanometer. Triglycerides, total cholesterol, high-density lipoprotein cholesterol in plasma were measured after at least a 9-hour overnight fast using North American Lipid Research Clinics methodology. Low-density lipoprotein cholesterol levels were calculated using the Friedewald equation. Plasma glucose concentrations at 1 h after a 50-g glucose load (ppGlucose) were measured among those without a previous diagnosis of diabetes. Diabetes was defined by the current use of insulin or oral hypoglycemic agents, or ppGlucose > 250 mg/dL. A 12-lead electrocardiogram was recorded. Information on the current use of medications was collected. Participants were interviewed by a physician who completed a medical history questionnaire that included questions about cardiovascular events and procedures.

### 2.7. Follow-up and Assessment of Endpoints

Vital status and the cause and date of death were ascertained through medical records in five active follow-up examinations (exam 2, 1981–1982; exam 3, 1986–1987; exam 4, 1995–1997; exam 5, 1999–2001; and exam 6 2001–2003) [22] and later on using death certificates or the National Death Index [12] through 31 December 2014. As previously described [9,12], physicians assigned corresponding International Classification of Diseases, Ninth Revision codes (ICD-9) for morbidity outcomes. Death certificates or the National Death Index coded to the ninth revision codes were obtained for decedents. The endpoint was death from all cardiovascular diseases (390–398, 402, 404, 410–438). Subjects were considered lost to follow-up if a death certificate or coding from the National Death Index could not be traced. The follow-up was terminated at the date of death, end of follow-up, or loss to follow-up, whichever occurred first.

### 2.8. Statistical Analysis

#### 2.8.1. Estimation of Peripheral Blood Leukocyte Composition

The post-normalized beta values were calculated using the 450 K array methylation data following the published method [16]. Beta values with associated detection *p*-values greater than 0.01 were set to missing [16]. Blood leukocyte composition was estimated for six blood leukocyte subtypes, including CD4+ T-cells, CD8+ T-cells, natural killer (NK) cells, B-cells, monocytes, and granulocytes, from the 450 K array methylation data using the “estimateCellCounts” function in the R package “minfi” [23].

#### 2.8.2. Identification of Signature Differentially Hydroxymethylated Regions

Differentially 5-hydroxymethylated genomic regions (DhMR) were investigated in discovery/exploratory/training (phase 1), validation (phase 2), and generalizability (phase 3) phases. Phases 1 and 2 was a standard training-validation set-up: we used a random split-sample method to randomly split 19 CVD-dMZ pairs into a training cohort/set consisting of 12 pairs (phase 1) and a validation cohort/set consisting of 7 pairs (phase 2) with the sample size selected a priori. There were no overlaps in twin pairs between training and validation cohorts. The phase 2 study was an internal validation of the phase 1 study in a different set of CVD-dMZ. A generalization cohort/set for phase 3 consisted of 19 discordant DZ twin pairs (CVD-dDZ). Co-twins of a DZ twin pair share 50% genes on average, while MZ co-twins share 100% genes. Details were described as follows.

In phase 1, we tested DhMRs at the group level using our improved analytic pipeline that was adapted from a published R package MethylAction (version 1.0) [24]. We ran MethylAction using a sliding window size of 50 bp, a fragment size of 200 bp, the minimum DhMR size of 150 bp, a join distance of 200 bp. Detailed statistical methods of MethylAction were published by Bhasin et al. [24]. To summarize, MethylAction started with a filtering stage to remove the noise. Windows that were below sample-specific cutoffs across all samples were removed. Reads were normalized based on library size using DESeq (version 1.8.3) [25] for each pairwise comparison with the negative binomial test [25]; a false discovery rate (FDR) was established for each number of reads, and windows with greater than or equal to the lowest level of reads with an FDR of less than 10% were considered to contain signal [24]. To generate candidate regions, windows with equivalent adjacent patterns (i.e., either hyper- or hypo-hydroxymethylated pattern comparing cases with controls and vice versa) and within a 200 bp gap distance were then joined to create a set of noise-free regions. In our study, 183,099 noise-free regions were generated through the filtering stage and used for the next DhMR analysis.

Differences in read counts in the noise-free regions between the case and control groups were tested using analysis of deviance (ANODEV) for the generalized linear model to detect DhMRs, which allows adjustment for variables [24]. This model enables adjustment for paired subject designs [24]. Details were described as follows.

We first only included the exposure (i.e., the group variable) without any covariates in the ANODEV-based generalized linear models (model 1). Thus, no covariates were adjusted in model 1. However, potential confounding from germline and environment shared between co-twins of a twin pair was adjusted because co-twins were naturally matched for them through our study design. To further control for potential confounding, we then fitted two sets of covariates as potential confounders into models 2 and 3, respectively. We selected the first set of covariates consisting of five of the six subtypes of blood leukocytes a priori. The sixth subtype of blood leukocytes, granulocytes, was excluded because the sum of the composition percentage of the six subtypes of blood leukocytes was 100%, and we needed to avoid the near-perfect collinearity [26]. We fitted this covariate set into model 2 to adjust for five blood leukocytes. The second set of covariates consisted of age and the five blood leukocytes selected for model 2. We fitted the second covariate set into the model (model 3) to adjust for age and blood leukocyte composition. The ANODEV *p*-values were adjusted using the Benjamini-Hochberg (B-H) procedure FDR to control for multiple testing [27]. We selected DNA regions that were directionally consistent and differentially hydroxymethylated before and after covariate adjustment at a B-H FDR q ≤ 0.01 to discover DhMRs and reduce potential false-positive DhMRs as the number of DhMRs from the discovery/exploratory/training analysis was very large.

In phase 2, we evaluated the DhMRs discovered from phase 1 in the validation set of 7 dMZ pairs to identify candidate DhMRs. To remove the noise, we implemented a filter that removed any DhMR with a nominal *p*-value > 0.01 from the differential expression analysis. Given the small sample size of CVD-dMZ in phase 2 relative to phase 1, we used a less stringent significance level to reduce potential type 2 error (i.e., false negative). The candidates that passed were then filtered for DhMRs. The filtered DhMRs had the same log_2_-fold-change direction as the corresponding DhMRs from phase 1 among 12 dMZ pairs. We performed principal component analysis (PCA) on the candidate DhMRs. The graph-based visual extraction method was used to select the two dimensions (i.e., principal components, PC) that best-separated cases and controls to remove the noise and for parsimony. The transformation matrix of the 2 selected dimensions (which includes weights for rotation, centering, and scaling) was used in all subsequent calculations of the 2 scores used as surrogate indicators of the overall candidate DhMR effect. HOMER (version 4.8) (http://homer.ucsd.edu/homer/, accessed on 28 November 2017) was used to annotate the significant DhMRs [28].

In phase 3, the phase 2 transformation matrix was applied to the dDZ pairs and generated the dDZ DhMRs scores. A binary linear classifier was used to separate dDZ case co-twins from their control co-twins. Multiple classifiers were tested; many of them failed due to multiple optimal solutions to separate case and control. To identify and select only one of the solutions, a single node neural network with a tanh sigmoid activation function was used to define a binary linear classifier using the two PCA scores as the input in the DZ. The R package “neuralnet” was used for this neural network analysis. The area under receiver operating characteristic curve (AUC) was estimated in the DZ. We selected the binary linear classifier with high sensitivity, AUC, and specificity with the priority to the sensitivity and the AUC. Then we evaluated the performance of this classifier in the MZ pairs and the combined cohort of MZ and DZ pairs separately. A model based on neural network analysis was developed to predict cardiovascular death. The mergePeaks function from HOMER [28] was used to detect overlap between DhMRs.

We performed the sensitivity analysis of DhMRs among a subset of 13 stringently defined CVD-dMZ pairs and another subset of 13 MZ pairs primarily defined discordant for coronary heart death out of the primarily defined 19 CVD-dMZ pairs. Both subsets had an overlap of the 8 MZ pairs. As the DhMRs results were similar to those from 19 CVD-dMZ pairs, we reported the results from the primarily defined twin pairs discordant for cardiovascular disease.

### 2.9. Bioinformatic Analysis

#### 2.9.1. Bioinformatic Visualization

R and Bioconductor packages were used to visualize the graph-based results. The PCA plots were created using the limma package (version 3.34.0) to show the visual separation of case co-twins from their control co-twins. The heatmaps were created to visualize hydroxymethylation and clustering using the ComplexHeatmap package (version 1.15.1). Volcano plots were created to visualize the statistical *p*-values versus fold-change using ggplot2 (version 1.15.1).

#### 2.9.2. Functional Enrichment Analysis

The significant DhMRs were annotated with nearby genes using HOMER [28]. The bioinformatic tool, Enrichr [29] (https://maayanlab.cloud/Enrichr/, accessed on 23 June 2020), was used to perform the comprehensive enrichment analysis for these genes. The gene ontology (GO) analysis [30] consisted of analysis of three categories: biological process, cellular component, molecular function. Kyoto Encyclopedia of Genes and Genomes (KEGG) [31]. WiKipathway [32] and PATHER pathway [33] enrichment analyses were performed to illustrate the genes’ functions in metabolic pathways, gene regulatory pathways, and signaling pathways [34]. The ClinVar 2019 [35], dbGaP [36], and DisGeNET [37] disease analyses were conducted to reveal human diseases and health in relation to these genes. The GWAS Catalog [38] and the U.K. Biobank GWAS v1 [39] were used to disclose diseases, disorders, and phenotypes related to these genes based on genome-wide association studies. The Rare Diseases Gene References into Function (GeneRIF) Gene lists [40], Rare Disease GeneRIF ARCHS4 predictions archive, Rare Disease AutoRIF Gene Lists, and Rare Disease AutoRIF ARCHS4 predictions archive were used to identify modern humans’ rare diseases associated with genes mapped to the DhMRs. A nominal *p*-value < 0.05 was used as statistical significance. The statistically significant results that were female-specific were removed. The statistically significant results with the top 5 smallest *p*-values were listed if there were more than five statistically significant results.

#### 2.9.3. DNA Motif Enrichment Analysis

HOMER [28] motif finding algorithm was used to identify enriched DNA motifs that determined which DNA-binding transcription factors control the transcription of a set of genes. HOMER matched de novo motifs to known motifs for transcription factors. The quality criteria to identify motifs included a Bonferroni corrected significant level of 0.00049 to control for multiple testing and the percentage of target sequences with motif greater than 5%.

## 3. Results

### 3.1. Characteristics of the Study Twin Pairs Discordant for Cardiovascular Death

Differences in age at death or at the last follow-up date (years) between controls and cases ranged from 3.5 to 23.8 years (mean ± SD: 10.3 ± 5.7 years; median (interquartile range (IQR) 10.0 (9.3) years)) for MZ pairs and 1.7– 17.7 years (mean ± SD: 8.49 ± 5.76 years; median (IQR) 7.5 (10.9) years) for DZ pairs. The identical case twins had a lower level of low-density lipoprotein (LDL) cholesterol at exam 3 than their control twins (*p* = 0.013), while the fraternal case twins tended to use more antihypertensives at exam 3 than their control twins (*p* = 0.08) (Appendix A). Peripheral blood leukocyte composition was not statistically significantly different between case twins and their co-twin controls (Appendix A).

### 3.2. Differentially Hydroxymethylated Regions (DhMRs) from Monozygotic (MZ) Twin Pairs Discordant for Cardiovascular Death (CVD-dMZ)

In phase 1, 1893 DhMRs (case twins versus control twins) were statistically significant and directionally consistent among three regression models without and with subsequent controlling for blood leukocyte composition and age (B-H false discovery rate (FDR) < 0.01) (Appendix A). In phase 2, a total of 102 DhMRs out of 1893 DhMRs were statistically significant at a nominal *p* < 0.01 and had the directional consistent log_2_-fold changes between the training and the validation cohorts. Figure 1 is the graph-based visualization of the 102 DhMRs. Figure 1a shows that two distinct clusters formed by the DNA regions where cases and controls differ in hydroxymethylation, of which differentially hyper-hydroxymethylated regions (hyper-DhMRs) are more predominant. Figure 1b illustrates that the majority of the 102 DhMRs have less than two-log_2_-fold changes; there are 16 hyper-DhMRs and 2 differentially hypo-hydroxymethylated regions (hypo-DhMRs) with log_2_-fold-change greater than two out of 102 DhMRs. The overall effects of 102 DhMRs were represented with the principal component score (Figure 1c). Visual inspection of the graphs shows distinct separation and pattern between case and control twins along dimension 1 (i.e., principal component score 1) (Figure 1c).

#### 3.2.1. Genetic Characteristics of the 102 DhMRs

Among 19 MZ pairs, these 102 DhMRs were located on 21 autosomal chromosomes and the X chromosome except for chromosome 21 or the Y chromosome (Appendix A). They were within/overlapped with exons, intergenic regions, introns, promoter transcription start sites (TSS), and transcription terminal sites (TTS) (Table 1). None of 102 DhMRs were within/overlapped with 3′-UTR, 5′-UTR, or non-coding regions. Among these DhMRs, 84 (82.4%) DhMRs were hyper-hydroxymethylated with log_2_-fold changes ranging from 0.73 to 3.64, (i.e., hyper-DhMRs) (Figure 1b) and predominantly with intergenic regions and introns (Table 1), while 18 (17.6%) of DhMRs were hypo-hydroxymethylated (i.e., hypo-DhMRs) with log_2_-fold changes ranging from −2.87 to −0.36 (Figure 1b) and predominantly with introns (Table 1). Figure 1c visualizes the separation of case twins from control twins.

#### 3.2.2. Generalizability Validation in Dizygotic (DZ) Twin Pairs Discordant for Cardiovascular Death (CVD-dDZ)

In phase 3, dimensions 1 and 4 from the 102 robust candidate DhMRs were used in the neural network analysis to validate DhMRs among 19 DZ twin pairs (Figure 2a,c). From the simulation, a binary linear classifier with the highest sensitivity and the area under the curve was identified (Figure 2a,b). The AUC was 0.72, indicating a suitable prediction for CVD deaths (Figure 2b). Appendix A details the sensitivity, specificity, positive predictive value, and negative predictive value of the classifier’s ability to predict the case status in the MZ, DZ, and combined cohort. Figure 2c illustrates that the linear classifier of the MZ pairs is a vertical line through the origin; the 2-step 2D transformation via topological changes from the MZ classifier to the DZ binary classifier with a slope of 0.214 is a clockwise rotation at the origin by 72.92° and then a movement upwards along the vertical axis by 0.664 units (Figure 2c), suggesting genetic influences on the 102 DhMRs to distinct case co-twins from their control co-twins since DZ twins share fewer genes than MZ twins.

### 3.3. Functional Enrichment Analysis of DhMRs

All DhMRs could work together in vivo to execute functions; so, they were analyzed together. These DhMRs involved various biological processes, molecular activities, cellular structures, and pathways (Figure 3a and Appendix A) and were manifested by a wide range of phenotypes, disorders, diseases (Figure 3b and Appendix A), and rare diseases (Figure 3c and Appendix A) (all nominal *p* < 0.05). As each subtype of DhMRs (i.e., hyper-DhMRs and hypo-DhMRs) could have different regulatory functions, DhMRs subtype-specific bioinformatic analyses were performed (Appendix A). There were no overlaps for pathways, phenotypes, disorders, diseases, and rare diseases between hyper-DhMRs and hypo-DhMRs (Appendix A). Representative pathways overrepresented for hypo-DhMRs were a couple of Wikipathways (i.e., G protein signaling calcium and regulation in cardiac cells) (multiple-testing adjusted *p_adj_* < 0.05) (Figure 3e).

Other potential pathways revealed at least twice were related to the negative regulation of potassium ion in the GO biological process, synapse regulation in the GO biological process and the KEGG pathway, and the p53 signaling in KEGG and Panther pathways linking all DhMRs and hyper-DhMRs to CVD deaths (Appendix A). Representative phenotypes overrepresented for hyper-DhMRs were two GWAS phenotypes, i.e., mean arterial pressure (alcohol consumption interaction) and adverse response to chemotherapy in breast cancer (alopecia), although breast cancer was rare among males (multiple-testing adjusted *p_adj_* < 0.05) (Figure 3d). Representative phenotypes/disorders/diseases overrepresented for hypo-DhMRs with multiple-testing adjusted *p_adj_* < 0.05 were three GWAS phenotypes (i.e., “waist circumference”, “body mass index in physically active individuals”, and “body mass index (joint analysis main effects and physical activity interaction)”) (Figure 3f), three ClinVar diseases (i.e., “familiar colorectal cancer”, “carcinoma of colon”, and “primary ciliary dyskinesia”) (Figure 3g), and one dbGap phenotype (i.e., “epilepsies, partial”) (Figure 3h).

### 3.4. Enriched DNA Motifs

Four, five, and two enriched DNA motifs were identified from genes mapped to all DhMRs, hyper-DhMRs, and hypo-DhMRs, respectively (Figure 4). Of the four de novo motifs for all DhMRs, two were matched for the transcription factor 7 (TCF7) binding site (*p* = 1.0 × 10^−9^ and *p* = 1.0 × 10^−6^). All the remained de novo motifs were uniquely matched for known motifs.

## 4. Discussion

Our study showed that the signature DhMRs for cardiovascular death in circulating leukocytes were independent of blood leukocytes composition, germline, environmental factors shared between co-twins (i.e., common environmental factors), and age and might be influenced by genes. The enrichment-based whole-genome DhMRs were predominantly hyper-DhMRs, which were mainly within or overlapped with intergenic regions and introns. Comprehensive bioinformatic analyses revealed potential pathophysiological mechanisms linking the DhMRs to cardiovascular death as well as phenotypes and diseases related to genes mapped to the DhMRs. We also computationally discovered three DNA sequence motifs as promising gene regulatory elements. In brief, our findings shed light on epigenetic mechanisms underlying cardiovascular death.

### 4.1. Consistency with Prior Studies

Previous population studies of circulating hydroxymethylation in relation to cardiovascular disease are few. Levels of global DNA 5hmC in peripheral blood mononuclear cells and aortic atherosclerotic tissue were higher in elderly patients with coronary heart disease than controls [7]. The same research group also found that the higher hydroxymethylation levels in peripheral blood mononuclear cells were associated with coronary atherosclerosis in a case-control study that included 91 carotid atherosclerotic patients [41]. Of these cases, 53 cases also had coronary heart disease, and 11 cases had stroke events [41]. Our study provides evidence on the signature hydroxymethylated regions of DNA in circulating leukocytes in which 5hmC levels were prospectively related to cardiovascular death, independent of blood leukocyte composition, germline, shared environment, and age. Our comprehensive bioinformatic analyses of genes mapped to the DhMRs implied the consistency with previous studies of genes in relation to diseases, including cardiovascular disease.

### 4.2. Mechanisms

Oxidative stress, inflammation, and endothelial dysfunction were well-known pathophysiological mechanisms underlying cardiovascular disease [42,43]. Thus, the potential pathophysiological mechanisms linking hydroxymethylation to cardiovascular disease were multifaceted.

Oxidized LDL is a biomarker of oxidative stress and plays a crucial role in atherosclerotic cardiovascular disease [44]. Three TET proteins are dioxidases and needed molecular oxygen as well as co-factors Fe^2+^ and α-ketoglutarate (αKG) to oxidize 5mC into 5hmC [45]. Tet2-mediated hydroxymethylation inhibits oxidized LDL-induced endothelial dysfunction [46]; therefore, TET2 protein is anti-atherosclerotic [47]. Moreover, TET enzymes and 5hmC affect adaptive and innate immune response [45] and thus play a pivotal role in inflammation toward cardiovascular disease. Previous laboratory cell experiments revealed the role of Tet proteins in genome-wide hydroxymethylation [48]. In mouse embryonic stem cells, Tet1 enzyme primarily regulated 5hmC levels at gene promoters and transcription start sites (TSS), whereas Tet2 enzyme mainly regulated 5hmC levels in gene bodies, exon boundaries of highly expressed genes, and exons, respectively [48]. TET3 enzyme acts on enhancer demethylation [45]. In our study, the number of hyper-DhMRs was two times higher than that of hypo-DhMRs. The percentage of hyper-DhMRs associated with introns was two-third of that of hypo-DhMRs. Of 84 hyper-DhMRs, half were related to intergenic regions that contained promoters and enhancers, one with TSS, one with exon, and the remainder of hyper-DhMRs with introns. It was hypothesized that all TET proteins might be involved in demethylation to generate the hyper-DhMRs. At the same time, the hypo-DhMRs could result from the suppressed demethylation catalyzed by all TET proteins, the reduction in 5mC due to deamination-mediated demethylation, or both.

The predominant significant DhMRs were hyper-DhMRs in our study. In contrast to 5-mC, 5-hmC increases DNA flexibility and enhances the mechanical stability of the nucleosome [49]. Therefore, the hyper-DhMRs in our study might improve the accessibility of DNA to various proteins, including chromatin regulators, transcription, and replication machineries [50]. By contrast, our hypo-DhMRs would reduce the accessibility of DNA to various proteins. Like previous studies using either all DhMRs [51] or DhMR subtypes [52], our comprehensive bioinformatic annotation of genes mapped to all, hyper-DhMRs, and hypo-DhMR with enriched ontology and pathway terms provided comprehensive biological processes, molecular functions, and pathways, which were potential pathophysiological mechanisms linking DhMRs to cardiovascular death risk. The G protein signaling calcium pathway and regulation in the cardiac cell pathway were the most promising pathophysiological mechanisms linking hypo-DhMRs to cardiovascular death risk. Negative regulation of potassium ion, synapse regulation, and p53 signaling pathways were potential pathophysiological mechanisms linking all DhMRs and hyper-DhMRs to cardiovascular deaths.

Our computational bioinformatic discovery of DNA motifs revealed candidate DNA transcriptional regulatory elements regulating gene expression through hydroxymethylation [53,54,55]. For example, the most promising discovered DNA motif in our study was matched to a known motif for Zinc Finger and BTB Domain Containing 18 (ZBTB18). *ZBTB18* (i.e., *ZNF238* or *RP58*) gene encodes a C2H2-type zinc-finger protein that acts as a transcriptional repressor of genes. ZNF238/RP58 regulates the myogenesis genome network [56], brain development, and brain functions [57]. Zinc-finger transcription factors prefer methylated cytosine in vitro [58]. Zinc-finger transcription factors prefer methylated cytosine in vitro [58]. Therefore, oxidation of methylated cytosine to hydroxymethylated cytosine might regulate *ZBTB18* gene expression via the reduction in the affinity of transcription factors to this gene in vivo [59].

Little is known about the mechanism linking the *ZBTB18* gene to cardiovascular disease among adults. ZBTB18 plays a crucial role in myogenesis by directly repressing the expression of the family of inhibitors of the DNA (ID) gene *ID* [60]. The postulated mechanism is that ZBTB18 regulates the gene expression for ID proteins (IDs), including ID1, ID2, ID3, and ID4. Gene polymorphisms are mechanistically similar to hydroxymethylation in the way to affect gene expression. Therefore, evidence about gene polymorphisms associated with cardiovascular disease in the population study can be parallel evidence, at least partially, to support our postulated mechanism linking *ZBTB18* to cardiovascular disease via IDs, particularly ID2 [61], ID3 [62,63,64], and ID4 [65]. No previous population studies reported an association of *ID1* gene polymorphism with cardiovascular disease. The low expression of the *ID* gene exists in most of the normal adult tissues [66]. At the cellular level, Id1, Id2, and Id3 induce activation and proliferation of endothelial cells [66,67]; ID2 and ID3 are involved in vascular cell differentiation [67], while ID1 and ID3 are involved in angiogenic processes [66,67]. IDs contribute to atherosclerosis through their involvement in the accumulation of macrophages in the vascular wall, oxidized LDL-mediated lipid accumulation and plaque formation, smooth muscle cell proliferation, thrombosis [67], and vascular calcification [68]. IDs are probably involved in angiogenesis-related atherosclerotic plaque growth and instability [69]. Therefore, our mechanistic hypothesis is that the hydroxymethylation of the ZBTB18 motif could regulate the expression of *ID*s and thus contribute to cardiovascular disease.

Taken together, our findings yielded putative mechanisms for further studies to investigate exact mechanisms.

### 4.3. Limitations and Advantages

There were limitations to our study. Our discordant twin sample size was limited on the surface, although it was the largest, the most extended follow-up, prospective discordant twin study in this kind of research. The root cause of this apparent small sample size was the extremely low twin birth rate in comparison with the singleton birth rate. The NHLBI Twin Study twins were born between 1917 and 1927, when the twinning birth rate was likely to be 1.36% [70,71,72]. The live births were roughly estimated as 28,605,000 during 1917–1927 in the U.S. [73]. Assuming 50% of live births were males, the live births of male twins would be 194,514, out of which roughly 1/3 were MZ, and 2/3 were DZ [72]. Therefore, the 254 MZ pairs and 260 DZ pairs enrolled in the NHLBI Twin Study entire cohort were equivalent to 110,535 and 56,573 male singletons, respectively. Such a sample size was much larger than that of 2344 men enrolled in the original cohort of Framingham Heart Study designed to investigate cardiovascular disease and its etiology [74]. Likewise, the sample size of our discordant twin pairs was large considering the twinning birth rate. Overfitting might be a concern in models adjusting for blood leukocyte composition. We used several methods to minimize overfitting. First, blood leukocyte subtypes were defined a priori as confounders and thus forced into the model since blood leukocyte composition was known as a confounder in the association of blood DNA methylation with phenotypes or outcomes in studies [26] and 5hmC is derived from 5mC. Second, we reported DhMRs unadjusted and adjusted for blood leukocyte composition and demonstrated that some DhMRs were not materially influenced by adjustment and were robust to overfitting. The robustness to overfitting was statistically plausible as overfitting affects strong signals less than moderate ones [75]. Biologically, a previous study supported the existence of a non-cell-mediated differential DNA methylation process [76]. As 5hmC is derived from 5mC, it is biologically plausible that some differential DNA hydroxymethylation was not mediated through cells. Taken together, we minimized overfitting by considering both biological/clinical and biostatistical importance. Hydroxymethylation enriched fragments obtained by the hMe-Seal method were not base pair-specific. It was biologically plausible that hydroxymethylation executed its function through its cluster in the specific DNA region. We did not measure messenger RNA in blood as the mRNA preservation technique was not used in the mid-1980s when the NHLBI Twin Study collected biospecimens. Thus, we were unable to assess the influence of DhMRs on gene transcription. We did not measure changes in gene expression of any gene related to pathways revealed from bioinformatic analysis. However, our research yielded evidence for future investigation of specific DNA regions at the base-pair resolution and the influence of hydroxymethylation on gene transcription and expression.

There were some advantages of our study. Our prospective design with discordant twin pairs was equivalent to the 1:1 individually matched, nested case-control design and thus might be called “nested case-co-twin-control design.” We specified the “later” periods as one year after the index date until the end of the follow-up [14,77]. The advantage of this definition was “to make inference under a proportional hazards model from the conditional logistic approach” [78]. Thus, this design could provide unbiased relative risk estimates [79]. Confounding could interfere with DhMRs toward cardiovascular death if confounders, confounding factors, were not distributed equally between cases and controls. However, we used the matching design and the statistical modeling to control for potential confounding to minimize potential confounding bias. Our discordant twin pair design was unique to naturally match case twins with their control twin brothers for germline and environmental factors shared between twins. The human genome consists of 3.2 billion base pairs (2% are exons) [80]. Our nested case-co-twin-control design was the optimal, natural, human experimental structure to control for confounding from germline and common environment, including age-cohort-period effects, family history of disease, blood sample collection, sample storage conditions, biochemical assay runs, and numerous other unknown or unmeasured common environmental factors [9,12]. We characterized that the signature DhMRs were predominantly within or overlapped with intergenic regions and introns as well as predominant with hyper-DhMRs over hypo-DhMRs independent of germline and shared environmental factors. As MZ twin pairs share 100% of the germline, our study warrants further investigations of the environmental origins of these signature regions in vivo and in vitro, which, in turn, can provide insights into preventive and therapeutic targets. Our study provides evidence to support the existence of a non-cell-mediated differential DNA hydroxymethylation process. By the inclusion of dizygotic twin pairs, we were able to demonstrate that genetic factors might play a role in the influence of hydroxymethylation on cardiovascular death. Our extended longitudinal design demonstrated that hydroxymethylation temporally occurred before cardiovascular death. This temporal order was critical to explain the causal role of hydroxymethylation in cardiovascular death.

## 5. Conclusions

In conclusion, we found signature DhMRs in circulating leukocytes associated with cardiovascular death prospectively, and their binary classifiers differed by zygosity, suggesting that hydroxymethylation is a pathophysiological mechanism underlying cardiovascular death and may be influenced by genetic factors.

## Figures and Tables

**Figure 1 genes-12-01183-f001:**
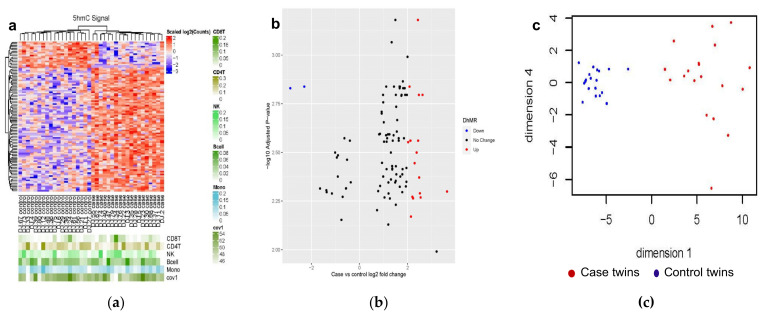
Graph-based visualization of the 102 candidate differentially hydroxymethylated regions (DhMRs) among 19 monozygotic twin pairs discordant for cardiovascular death. (**a**). Heatmap. Abbreviation: CD8T = CD8+ T-cells; CD4T = CD4+ T-cells; NK = natural killer cells; Bcel = B-cells; Mono = monocytes; Cov1 = age; 5hmC = 5-hydroxymethylcytosine. Each row represents a DhMR, and each column represents a participant. The normalized count values are log_2_ transformed and then scaled with z-score in the blue-red color scale. The 102 DhMRs were adjusted for leukocyte subtypes and age. The lower level of hydroxymethylation among co-twins than their twin brothers is in blue, and the higher level of hydroxymethylation than their twin brothers is in red; (**b**). volcano plot. The horizontal axis represents the log_2_ fold-change (FC) (case versus control twins), and the vertical axis corresponds to the negative log_10_ (adjusted *p*-value). Red dots represent the significant (−log_10_ adjusted *p*-value > 2.0) hyper-DhMRs (log_2_ FC > 2) and blue dots represent significant (−log_10_ adjusted *p*-value > 2.0) hypo-DhMRs (log_2_ FC < −2) when comparing cases with their corresponding control co-twins; and (**c**). scatter plot. The overall effect of the 102 DhMRs was represented with principal components (PC) 1 and 4 from principal component analysis. The horizontal axis represents PC1 (i.e., dimension 1), and the vertical axis displays PC4 (i.e., dimension 4). Red dots represent case twins, while blue dots represent their control twin brothers.

**Figure 2 genes-12-01183-f002:**
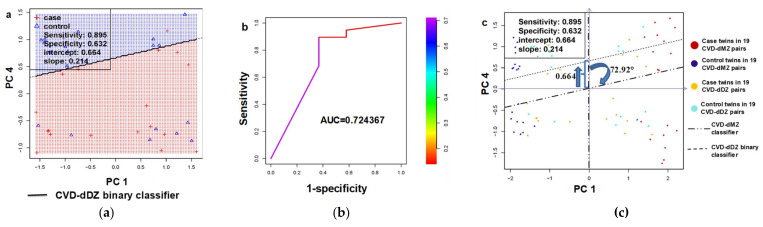
Visualization of the binary linear classifier among 19 dizygotic twin pairs (DZ) discordant for cardiovascular death (CVD-dDZ) generated with the 102 candidate DhMRs identified from monozygotic twin pairs (MZ) discordant for cardiovascular death (CVD-dMZ) using 1000 times simulation. (**a**) A principal component analysis of the 102 DhMRs generated principal component 1 (PC1) and principal component 4 (PC4). The points in the area below the binary linear classifier are case twins while points above the classifier are control twins. Binary linear classifier PC4 3 =0.664  0.214×PC1 3; (**b**) receiver operating characteristic (ROC) curve and the area under the curve (AUC) using the binary linear classifier for the CVD-dDZ; (**c**) graph-based 2D topology of genetic influences on the binary linear classifier by comparison between MZ and DZ pairs. Two blues arrows show the two-step topological changes (i.e., 2-D transformation) of the MZ classifier (i.e., the vertical line across the origin) to the DZ binary classifier line with a slope of 0.214. The first step is indicated with a clockwise half-circle arrow representing the clockwise rotation of the MZ classifier at the origin by 72.92°. The second step is indicated with an upward arrow in blue representing an upward movement of the rotated MZ classifier along the vertical axis by 0.664 units.

**Figure 3 genes-12-01183-f003:**
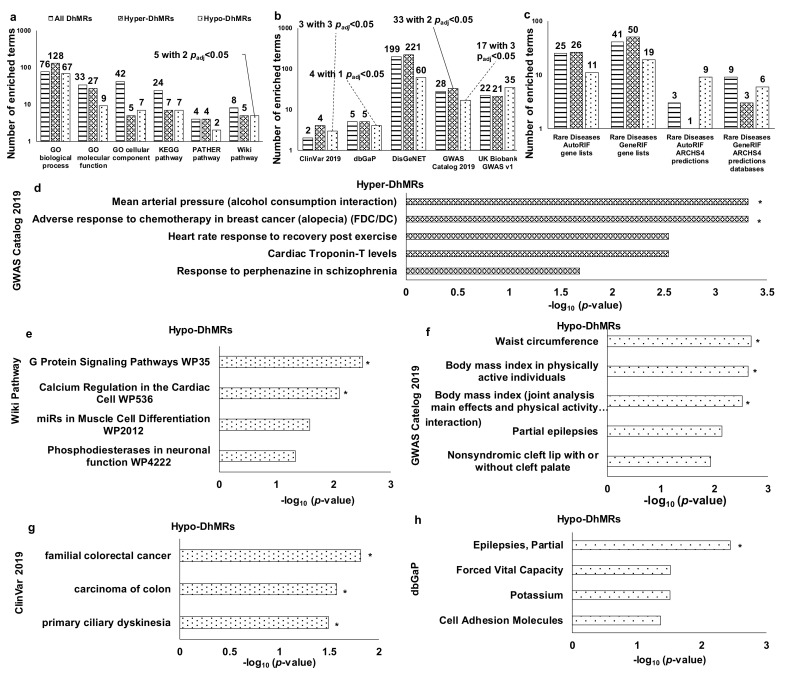
Biological functions and pathways, phenotypes, disorders, and (rare) diseases apparently relevant to males. (**a**) The number of enriched terms for biological functions and pathways (*p* < 0.05); (**b**) the number of enriched terms for phenotypes, disorders, diseases and rare diseases (*p* < 0.05); (**c**) the number of enriched terms for rare diseases (*p* < 0.05); (**d**–**h**) up to top 5 enriched terms with the smallest nominal *p*-values < 0.05 for the top terms containing any terms with *p*_adj_ < 0.05. * *p*_adj_: multiple-testing adjusted *p*-value using the Benjamini-Hochberg False Discovery Rate (FDR) procedure.

**Figure 4 genes-12-01183-f004:**
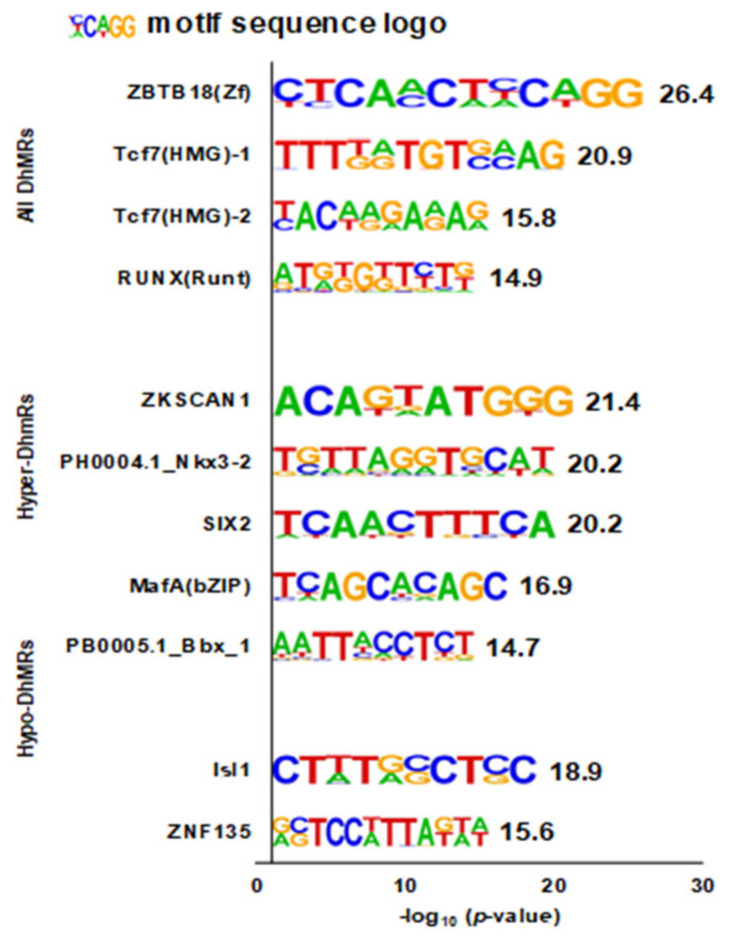
De novo motif identification from genes mapped to differentially hydroxymethylated regions (DhMRs) with HOMER. Abbreviations: DhMRs = differentially hydroxymethylated regions; hyper-DhMRs = differentially hyper-hydroxymethylated regions (case vs. control twins); hypo-DhMRs = differentially hypo-hydroxymethylated regions (case vs. control twins). The value next to the motif sequence logo is the negative log_10_ (*p*-value).

**Table 1 genes-12-01183-t001:** Annotation of the nearest genes mapped to the 102 differentially hydroxymethylated regions.

Annotation ^1^	Total DhMRs	Hyper-DhMRs	Hypo-DhMRs
n	102	84	18
Exons^1^	2 (2.0 ^3^)	2 (2.3 ^3^)	0
ncRNA^2^	1 (50 ^4^)	1 (50 ^4^)	0
protein-coding^2^	1 (50 ^4^)	1 (50 ^4^)	0
Pseudo^2^	0	0	0
snoRNA^2^	0	0	0
Intergenic regions^1^	47 (46 ^3^)	42 (50 ^3^)	5 (28 ^3^)
ncRNA	14 (30 ^4^)	13 (31 ^4^)	1 (20 ^4^)
protein-coding	30 (64 ^4^)	26 (62 ^4^)	4 (80 ^4^)
pseudo	3 (6 ^4^)	3 (7 ^4^)	0
snoRNA	0	0	0
Introns^1^	49 (48 ^3^)	37 (44 ^3^)	12 (67 ^3^)
ncRNA	11 (22 ^4^)	10 (27 ^4^)	1 (8 ^4^)
protein-coding	35 (71 ^4^)	25 (68 ^4^)	10 (83 ^4^)
pseudo	3 (6 ^4^)	1 (3 ^4^)	1 (8 ^4^)
snoRNA	1 (2 ^4^)	1 (3 ^4^)	0
TSS^1^	2 (2.0 ^3^)	1 (1.2 ^3^)	1 (5.6 ^3^)
ncRNA	0	0	0
protein-coding	1 (50 ^4^)	0	1 (100 ^4^)
pseudo	1 (50 ^4^)	1 (100 ^4^)	0
snoRNA	0	0	0
TTS ^1^	2 (2.0 ^3^)	2 (2.3 ^3^)	0
ncRNA	0	0	0
protein-coding	2 (100 ^4^)	2 (100 ^4^)	0
pseudo	0	0	0
snoRNA	0	0	0

Abbreviations: TSS = promoter transcription start sites; TTS = transcription terminal site; ncRNA = non-coding RNA; DhMRs = differentially hydroxymethylated regions; hyper-DhMRs = differentially hyper-hydroxymethylated regions (case twins vs. control twins); hypo-DhMRs = differentially hypo-hydroxymethylated regions (case twins vs. control twins). Data are presented as n (%) unless otherwise specified. ^1^ annotation, while 3′-UTR, 5′-UTR, and non-coding regions are not shown as none of the nearest genes, are within/overlapped with them; ^2^ gene type; ^3^ percentage of each annotation for all DhMRs and each type of DhMRs, respectively; ^4^ percentage of each gene type per annotation.

## Data Availability

The data from this study will be submitted to the NCBI Gene Expression Omnibus (GEO; https://www.ncbi.nlm.nih.gov/geo/) upon the acceptance of our manuscript. All relevant data supporting the key findings of this study are available within the article and the Appendix A.

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
