# Peer review of "Whole-Genome Differentially Hydroxymethylated DNA Regions among Twins Discordant for Cardiovascular Death"

_genes, 2021, doi:10.3390/genes12081183_

Round 1

Reviewer 1 Report

The manuscript "Whole-Genome Differetially Hydroxymethylated DNA Regions among Twins Discordant for Cardiovascular death"  find several regions differentially hydroxymethylated in discordant twins. Results are clear and the manuscript is well organized. The article is really intersting in the field. However I have some minor points:

  • Abstract (line 3-4): the sentence "Male twin pairs... 2014 included" is not well writen, it should be changed  to "Male twin pairs... 2014 were included"
  • Introduction: this section is a bit poor. Authors may add more information about 5hmC in cardiovascular diseases
  • Results and discussion: authors have identified different pathways in Hyper-DhMRs and hypo-DhMRs. There are changes in gene expression of any gene related with   these pathways?

Author Response

Dear Reviewer:

We are grateful to the reviewer for the constructive comments. We have addressed the reviewer's comments in the point-by-point manner as follows and believe the manuscript's clarity to be much improved.

The line number in the revised manuscript is different from that in the initial manuscript due to the revision. Therefore, in our response to the reviewer's comment, we use the line number in the revised manuscript with change tracking, entitled “genes-1314486_1st_round_revision_track_change_final.”

Below are our point-by-point responses:

  1. Abstract (line 3-4): the sentence "Male twin pairs... 2014 included" is not well writen, it should be changed  to "Male twin pairs... 2014 were included"

Response: Changes were made in the "Abstract" section (line 31).

  1. Introduction: this section is a bit poor. Authors may add more information about 5hmC in cardiovascular diseases

Response: Previous human studies of hydroxymethylation and cardiovascular hard outcomes were few. After searching PubMed for studies of hydroxymethylation and cardiovascular hard outcome, we were able to add one more human study (lines 58-59 and 62-65) to address this concern. In addition to human studies, we also added an animal experiment to address hydroxymethylation in heart tissue in response to another reviewer’s comment in lines 56-58.

  1. Results and discussion: authors have identified different pathways in Hyper-DhMRs and hypo-DhMRs. There are changes in gene expression of any gene related with   these pathways?

Response: We agree that the experiment on gene expression is important. As editors give us only seven days to revise the manuscript, we are unlikely to perform the gene expression experiment additionally. Therefore, we address this comment as a limitation in lines 645-646 and 648 in the "Discussion" section. We will do the gene expression experiment in the future when the condition is available.

Reviewer 2 Report

In this study, the authors examine whether DNA hydroxymethylation is prospectively associated with the risk for cardiovascular death independent of germline and common environment. To do this, they use a twin cohort study and a well-established method to measure genome-wide DNA hydroxymethylation – hMe-Seal-sequencing. This manuscript uses appropriate statistical methods to test their hypothesis, and identified a number of differentially hydroxymethylated regions throughout the genome in cardiovascular disease cases vs. controls. The potential role for DNA hydroxymethylation in regulating disease states is gaining steam in the literature, and this manuscript uses a unique twin cohort to prospectively examine this epigenetic mark’s association with known cardiovascular disease. While the hypothesis being tested in this paper is an important question for the field, this paper is hampered by a number of issues related to clarity of writing, data presentation, and discussion. The statistical grounding for their large multivariate regression modeling is also difficult to justify given the small sample size. The results are intriguing, but a number of points need to be adjusted prior to publication.

Comments:

There are a number of grammatical mistakes throughout the manuscript. Please proofread and fix.

Throughout the manuscript, the authors use the past tense when describing the existing literature as well as the knowledge gaps surrounding epigenetics and cardiovascular disease. Given that existing data represent the current state of scientific knowledge, they should be in the present tense. As specific examples, this first occurs in the first two sentences of the abstract, and then continues throughout the paper – i.e. when describing the case-control study design or using “known” instead of “knowing” in sections 2.4 and 2.5. It also occurs a number of times in the discussion. Please address the past tense usage prior to publication.

LINE 29: “Included” is at the end of the sentence. This should come earlier in this sentence.

LINES 47-51: The assertions made in this introductory paragraph are too general. Simply saying that gene expression plays a role in development of cardiovascular disease provides little usable context for the reader, as the broad term “gene expression” will apply to any and all disease states. What type of genes are involved, and how? Furthermore, the statement “it is known that genes can also affect epigenetic modifications” doesn’t introduce anything specific to the study at hand. Which genes and which epigenetic modifications? The authors need to improve the context of their introduction.

METHODS: What is the genomic coverage like for the hMe-Seal method? Is it biased in any way by the enzymatic treatment? Please include brief details regarding these questions in the methods section.

LINES 212-213: The authors state: “All windows with either zero or below sample-specific cutoffs were removed.” Do they mean that all windows with either zero or below sample-specific cutoffs in a single sample were removed from that sample? Or that those windows were removed from all samples based on a zero value in a single sample? Please clarify.

LINES 221-236: The authors first describe a method for detecting DhMRs using and analysis of deviance (ANODEV) generalized linear model. However, they then go on to say that they used linear regression within the MethylAction R package to screen DhMRs. It is unclear from reading this methods sections how the results from these two separate analyses relate to each other. Are they part of a shared pipeline, with the results from the ANODEV being input into the MethylAction pipeline? Or were they separate analyses? And if the latter, these analyses appear redundant, so why do both?

LINES 230-233: The authors state: “Second, we adjusted for blood leukocyte compositions by inclusion of five of the six blood leukocytes, except for granulocytes, selected as a priori as covariates [23] to be adjusted in the statistical model to avoid the near perfect collinearity (model 2).” Given the small sample size in this study, it does not seem valid to include five covariates that provide similar information in the modeling. Generally, at least 10 samples per group are expected per term in a model, so including 5 covariates to adjust for cell type composition is likely not valid, and may be leading to false positives. The authors already bring up the potential issue of collinearity, so they appear familiar with this potetial issue, but then do little to address it apart from removing granulocytes. A suggested alternative approach would be the opposite – to only include the granulocytes variable. That way, the authors can correct for cell type composition in a simpler way while avoiding overfitting for their small sample size.

LINES 237-260: The methods outlined for “phase 2” and “phase 3”, while interesting, does not feel hypothesis driven. Rather, it appears that the authors arbitrarily selected a subset of samples to run through several statistical tools, and then attempted to use their results from this smaller analysis to train another tool to detect differential hydroxymethylation, which is what their previous regression analyses appears to already be doing. The authors either need to better explain these methods or highlight the value added from this follow-up work in a clearer way. Why not simply combine all the available data together and run a joint regression model? What does the neural network add? As currently written, it feels like data massaging to find more significance.

FIGURE 1: There is no description of Figure 1c in the figure caption. Please provide this information in the caption.

LINES 311-318: The authors identify 1,892 DhMRs with FDR<0.01, but then go on to focus on those 102 DhMRs that showed nominal significance in their phase 2 cohort. It is unclear why the authors decide to focus on only these 102 DhMRs out of the larger list when their new cutoff is nominal significance. There is a much larger set of potentially interesting and significant hits from their phase 1 study that they seem to ignore based on a smaller, less significant follow-up analysis, which seems counterintuitive. Please provide justification for this approach.

LINES 321-324: Why do the authors decide to focus on log2-fold change > 2? Even small changes in epigenetic marks have been shown to induce clear effects on gene expression. Please provide justification for this cutoff.

Table 1: Column 1 (which provides descriptive names for the categories) is difficult to read in its current format. Please adjust the formatting to make the larger categories (Introns, TSS, etc.) more apparent for the reader.

Figure 2a: The authors state for Figure 2a: “The points in the area below the binary linear classifier are case twins while points above the classifier are control twins.” However, in the actual plot, there appear to be a number of controls below the line and at least two cases above the line. As such, it is unclear how effective this PCA is at delineating the two groups, which throws the linear classifier into question.

Figure 2c: What do the two arrows in Figure 2c refer to?

LINE 373: There should be a “so” after the comma between “functions” and “they.”

LINE 417: It is unclear what the authors mean by “…and may be influenced by genetic factors.” Please clarify this language. What genetic factors do the authors mean here? Genomic region?

LINE 427: Need a “)” somewhere in this sentence.

LINES 434-435: The authors state that their results are “independent of…shared environment.” However, shared environment was controlled for through the twin-pair study design, not during modeling. As such, the statement that the results are independent of a shared environment is not accurate. In fact, they are dependent on a shared environment, since that is the context that the authors argue a twin-pair design provides. Please clarify this language.

LINES 443-462: The authors go into great detail discussing the potential role for Tet enzymes in both cardiovascular disease and DNA hydroxymethylation, which is a very interesting potential mechanistic link between their differential DNA hydroxymethylation and a disease phenotype. They go on to say “It was implied that all TET proteins might be involved in demethylation to generate the hyper DhMRs.” However, there is no data in the manuscript showing that Tet enzymes were modified in blood leukocytes between cases and controls. As such, the authors should temper their language by clarifying that they hypothesize that TET enzymes might be involved in demethylation.”

LINES 472-473: The authors state: “The annotated cellular components shed light on the elements involved in these mechanisms.” It is unclear what information this sentence provides. What are the cellular components, elements, and mechanisms noted here? Please provide additional context for readers regarding specific pathways and their functions, as well as how they might play a role in cardiovascular disease.

LINES 480-482: The authors state: “Therefore, oxidation of methylated cytosine to hydroxymethylated cytosine might regulate ZBTB18 gene expression via the reduction in the affinity of transcription factors to this gene in vivo.” However, they do not go on to say what this gene does, and how it might play a role in their disease of interest. Is ZBTB18-mediated transcriptional repression known to play a role in cardiovascular disease? If not, then why is this candidate regulatory element of interest?

LINES 503-506: “Second, we reported DhMRs unadjusted and adjusted for blood leukocyte composition and demonstrated that some DhMRs were not materially influenced by adjustment and robust to overfitting at a ratio of 7.6 for the sample size to the number of predictors.” The authors were clearly being very thoughtful about this, which is great to see. However, as they note, only “some” DhMRs were robust to the number of predictors, so concerns of overfitting remain. See previous discussion about limiting number of cell type variables to just one. Additionally, the authors go on to say: “The robustness to overfitting was statistically plausible as overfitting affects strong signals less than moderate ones.” This feels a bit disingenuous given the modest sample size; with so few samples per group, a single outlier could shift variance quite a bit, effectively making what the authors claim are moderate effects seem much bigger than they really are. Again, the authors may want to consider limiting the number of terms in their model.

LINE 515: “messager” should read “messenger.”

DISCUSSION: Generally, blood has been shown to have very low levels of DNA hydroxymethylation (see: https://www.ncbi.nlm.nih.gov/pmc/articles/PMC3290782/). Given these low levels compared to other tissues, would the authors expect their detected differential DNA hydroxymethylation to carry over to other tissues, like the heart?

Author Response

Dear Reviewer:

We are grateful to the reviewer for the constructive comments. We have addressed the reviewer's comments in the point-by-point manner and believe the manuscript's clarity is much improved.

The line number in the revised manuscript is different from that in the initial manuscript due to the revision. Therefore, in our response to the reviewers' comment, we use the line number in the revised manuscript with change tracking, entitled “genes-1314486_1st_round_revision_track_change_final.”

Below are our point-by-point responses.

  1. There are a number of grammatical mistakes throughout the manuscript. Please proofread and fix.

Response: We did the proofreading and made changes to address this concern throughout the manuscript.

  1. Throughout the manuscript, the authors use the past tense when describing the existing literature as well as the knowledge gaps surrounding epigenetics and cardiovascular disease. Given that existing data represent the current state of scientific knowledge, they should be in the present tense. As specific examples, this first occurs in the first two sentences of the abstract and then continues throughout the paper – i.e. when describing the case-control study design or using "known" instead of "knowing" in sections 2.4 and 2.5. It also occurs a number of times in the discussion. Please address the past tense usage prior to publication.

Response: Changes were made accordingly.

  1. LINE 29: "Included" is at the end of the sentence. This should come earlier in this sentence.

Response: A change was made in line 31.  

  1. LINES 47-51: The assertions made in this introductory paragraph are too general. Simply saying that gene expression plays a role in development of cardiovascular disease provides little usable context for the reader, as the broad term "gene expression" will apply to any and all disease states. What type of genes are involved, and how? Furthermore, the statement "it is known that genes can also affect epigenetic modifications" doesn’t introduce anything specific to the study at hand. Which genes and which epigenetic modifications? The authors need to improve the context of their introduction.

Response: Changes were made by adding more information and emphasizing the need to control for potential genetic and environmental confounding and the importance of the need to have a prospective study in lines 56-58 and 62-65.

  1. METHODS: What is the genomic coverage like for the hMe-Seal method? Is it biased in any way by the enzymatic treatment? Please include brief details regarding these questions in the methods section.

Response: The hMe-Seal provides genome-wide coverage to detect 5hmC signals. It has been widely used to profile 5hmC in different tissues and cell types [1,2]. The earlier works have demonstrated that there is no bias with the-Seal, and it is more sensitive than the antibody-based immunoprecipitation approach [1,3,4]. Changes were made in lines146-149.

  1. LINES 212-213: The authors state: “All windows with either zero or below sample-specific cutoffs were removed.” Do they mean that all windows with either zero or below sample-specific cutoffs in a single sample were removed from that sample? Or that those windows were removed from all samples based on a zero value in a single sample? Please clarify.

Response: As stated in a previous publication from the developers of the R package “MethylAction,”[5] “These read counts are filtered to produce a set of windows deemed to contain signal in at least one sample by removing all windows with either all zero counts or all counts below sample-specific cutoffs." Changes were made in lines 241-242.

  1. LINES 221-236: The authors first describe a method for detecting DhMRs using and analysis of deviance (ANODEV) generalized linear model. However, they then go on to say that they used linear regression within the MethylAction R package to screen DhMRs. It is unclear from reading this methods sections how the results from these two separate analyses relate to each other. Are they part of a shared pipeline, with the results from the ANODEV being input into the MethylAction pipeline? Or were they separate analyses? And if the latter, these analyses appear redundant, so why do both?

Response: The ANODEV-based generalized linear model is used as the linear regression model for models without and with adjustment for covariates. For better clarity, we made changes in lines 258-280.

  1. LINES 230-233: The authors state: “Second, we adjusted for blood leukocyte compositions by the inclusion of five of the six blood leukocytes, except for granulocytes, selected as a priori as covariates [23] to be adjusted in the statistical model to avoid the near perfect collinearity (model 2).” Given the small sample size in this study, it does not seem valid to include five covariates that provide similar information in the modeling. Generally, at least 10 samples per group are expected per term in a model, so including 5 covariates to adjust for cell type composition is likely not valid, and may be leading to false positives. The authors already bring up the potential issue of collinearity, so they appear familiar with this potetial issue, but then do little to address it apart from removing granulocytes. A suggested alternative approach would be the opposite – to only include the granulocytes variable. That way, the authors can correct for cell type composition in a simpler way while avoiding overfitting for their small sample size.

Response: As editors gave us only seven days to revise the manuscript, we were unlikely to perform additional analysis. Therefore, we addressed this overfitting concern as a limitation in lines 625-638 in the “Discussion” section. However, we will do additional analysis in the future when the condition is available.

  1. LINES 237-260: The methods outlined for “phase 2” and “phase 3”, while interesting, does not feel hypothesis driven. Rather, it appears that the authors arbitrarily selected a subset of samples to run through several statistical tools, and then attempted to use their results from this smaller analysis to train another tool to detect differential hydroxymethylation, which is what their previous regression analyses appears to already be doing. The authors either need to better explain these methods or highlight the value added from this follow-up work in a clearer way. Why not simply combine all the available data together and run a joint regression model? What does the neural network add? As currently written, it feels like data massaging to find more significance.

Response: Validation is crucial in epidemiological studies, including our discordant twin pair study. Phases 1 and 2 were a standard training-validation set-up: we used the random split-sample validation in phases 1 and 2 with CVD-dMZ twin pairs. Two methodological benefits of the split-sample method compared to the use of all available data together (i.e., a single-step approach) are learning from the data and reducing type 1 error (i.e., false positive). In addition, we used the generalizability validation in phase 3 with CVD-dDZ twin pairs to assess the generalizability of findings from MZ to DZ since the degree to which co-twins share genes are different between MZ and DZ. The neural network analysis demonstrated that the results from phase 2 were validated in dizygotic twins with the 2-D transformation. The need for the 2-D transformation may be attributed to that DZ twins differ on average 50% of genes between each other. Therefore, to improve clarity, we renamed the three phases and made other changes in lines 220-228, 258-261, and 438-440.

  1. FIGURE 1: There is no description of Figure 1c in the figure caption. Please provide this information in the caption.

Response: Changes were made in response to this comment in Figure 1.

  1. LINES 311-318: The authors identify 1,892 DhMRs with FDR<0.01, but then go on to focus on those 102 DhMRs that showed nominal significance in their phase 2 cohort. It is unclear why the authors decide to focus on only these 102 DhMRs out of the larger list when their new cutoff is nominal significance. There is a much larger set of potentially interesting and significant hits from their phase 1 study that they seem to ignore based on a smaller, less significant follow-up analysis, which seems counterintuitive. Please provide justification for this approach.

Response: As our response to this reviewer’s comment 9, validation is crucial in epidemiological studies, including our discordant twin pair study. We used the standard random split-sample validation in phases 1 and 2 with CVD-dMZ twin pairs. Two methodological benefits of the split-sample method compared to the use of all available data are learning from the data and reducing type 1 error (i.e., false positive). We used a stringent significance level in phase I to reduce potential false positive DhMRs as the number of DhMRs from the discovery/exploratory/training analysis was very large. In phase II, as the sample size of CVD-dMZ was relatively small, we used a less stringent significance level to reduce type 2 error (i.e., false negative). Thus, justification was added to improve clarity in lines 277-280 and 294-296.

  1. LINES 321-324: Why do the authors decide to focus on log2-fold change > 2? Even small changes in epigenetic marks have been shown to induce clear effects on gene expression. Please provide justification for this cutoff.

Response: Our results show that the majority of the 102 DhMRs have less than two log2-fold changes in lines 383-384. We mentioned the two log2-fold changes since this cutoff was commonly used in basic research publications. The change was made in response to this comment in lines 385-386.

  1. Table 1: Column 1 (which provides descriptive names for the categories) is difficult to read in its current format. Please adjust the formatting to make the larger categories (Introns, TSS, etc.) more apparent for the reader.

Response: Since each annotation is genetically meaningful, in table 1, we removed 3’-UTR, 5’-UTR, and non-coding region annotation and made other changes to improve the clarity.

  1. Figure 2a: The authors state for Figure 2a: “The points in the area below the binary linear classifier are case twins while points above the classifier are control twins.” However, in the actual plot, there appear to be a number of controls below the line and at least two cases above the line. As such, it is unclear how effective this PCA is at delineating the two groups, which throws the linear classifier into question.

Response: In Figure 2b, the area under the curve (AUC) of the receiver operating characteristic curve was greater than 0.7, indicating that the performance of the binary classifier was good [6]. This AUC (i.e., c-statistic) was comparable to using the well-established Framingham Risk Score to predict the 10-year risk for coronary heart disease.[7] Changes were made in lines 431-432.

  1. Figure 2c: What do the two arrows in Figure 2c refer to?

Response: Two blues arrows show the two-step topological changes (i.e., 2-D transformation) of the MZ classifier (i.e., the vertical line across the origin) to the DZ binary classifier line with a slope of 0.214. The first step is indicated with a clockwise half-circle arrow representing the clockwise rotation of the MZ classifier at the origin by 72.92°. The second step is shown with an upward arrow in blue representing an upward movement of the rotated MZ classifier along the vertical axis by 0.664 units. Changes were made in lines 435-436 and the Figure 2 (c) legend.

  1. LINE 373: There should be a “so” after the comma between “functions” and “they.”

Response: Changes were made accordingly (line 458 in the revised manuscript).

  1. LINE 417: It is unclear what the authors mean by “…and may be influenced by genetic factors.” Please clarify this language. What genetic factors do the authors mean here? Genomic region?

Response: The DZ binary classifier was transformed from the MZ binary classifier through the 2-D topological changes. As the MZ twins share 100% genes and DZ twins share on average 50% genes, the 2-D topological changes may be attributable to, on average, 50% genes not shared between DZ twins. Therefore, to improve clarity, we changed “genetic factors” to “genes” in line 510.

  1. LINE 427: Need a “)” somewhere in this sentence.

Response: A change was made by deleting a “(” in the referred sentence.

  1. LINES 434-435: The authors state that their results are “independent of…shared environment.” However, shared environment was controlled for through the twin-pair study design, not during modeling. As such, the statement that the results are independent of a shared environment is not accurate. In fact, they are dependent on a shared environment, since that is the context that the authors argue a twin-pair design provides. Please clarify this language.

Response: Confounding could interfere with DhMR towards cardiovascular death only if confounders are not distributed equally between case and control groups. However, we used the matching design and the statistical modeling to control for potential confounding to minimize potential confounding bias. Confounding factors are called confounders. Matching factors are confounders. Both genes and environment shared between co-twins of a twin pair are not different from each other. So, case and control twins are naturally matched for them. As a result, we removed their confounding effect through the standard epidemiological matching. Moreover, we further removed confounding from confounders (age and leukocyte composition) by including them in the statistical model as covariates. It is a common epidemiological practice to use the “independent of confounders” verbiage if the confounders are controlled. Thus, our DhMRs were independent of common germline, shared environment, age, and leukocyte composition. To improve clarity, we made changes in lines 265-267 and 656-662

  1. LINES 443-462: The authors go into great detail discussing the potential role for Tet enzymes in both cardiovascular disease and DNA hydroxymethylation, which is a very interesting potential mechanistic link between their differential DNA hydroxymethylation and a disease phenotype. They go on to say “It was implied that all TET proteins might be involved in demethylation to generate the hyper DhMRs.” However, there is no data in the manuscript showing that Tet enzymes were modified in blood leukocytes between cases and controls. As such, the authors should temper their language by clarifying that they hypothesize that TET enzymes might be involved in demethylation.”

Response: Following this reviewer’s guidance, we made changes in line 553 to tone down the statement.

  1. LINES 472-473: The authors state: “The annotated cellular components shed light on the elements involved in these mechanisms.” It is unclear what information this sentence provides. What are the cellular components, elements, and mechanisms noted here? Please provide additional context for readers regarding specific pathways and their functions, as well as how they might play a role in cardiovascular disease.

Response: Changes were made in lines 478-481 and 568-572.

  1. LINES 480-482: The authors state: “Therefore, oxidation of methylated cytosine to hydroxymethylated cytosine might regulate ZBTB18 gene expression via the reduction in the affinity of transcription factors to this gene in vivo.” However, they do not go on to say what this gene does, and how it might play a role in their disease of interest. Is ZBTB18-mediated transcriptional repression known to play a role in cardiovascular disease? If not, then why is this candidate regulatory element of interest?

Response: We added information in lines 581-583 and 588-607 in response to this comment.

  1. LINES 503-506: “Second, we reported DhMRs unadjusted and adjusted for blood leukocyte composition and demonstrated that some DhMRs were not materially influenced by adjustment and robust to overfitting at a ratio of 7.6 for the sample size to the number of predictors.” The authors were clearly being very thoughtful about this, which is great to see. However, as they note, only “some” DhMRs were robust to the number of predictors, so concerns of overfitting remain. See previous discussion about limiting number of cell type variables to just one. Additionally, the authors go on to say: “The robustness to overfitting was statistically plausible as overfitting affects strong signals less than moderate ones.” This feels a bit disingenuous given the modest sample size; with so few samples per group, a single outlier could shift variance quite a bit, effectively making what the authors claim are moderate effects seem much bigger than they really are. Again, the authors may want to consider limiting the number of terms in their model.

Response: The epidemiologically specific meaning to control for five covariates is different from controlling for one covariate. For example, by controlling for five cell subtypes, we interpreted it as the distribution of each of five cell types was equally between cases and control twins. This interpretation is biologically different from a saying, “the distribution of one cell type was equally between cases and control twins through controlling for one cell type.” Therefore, biologically/clinically, it was essential to control five rather than one cell type. In addition, we are given only seven days by editors to revise the manuscript; it is unlikely for us to perform the suggested analysis additionally. To address this comment, we changed “overfitting was not a concern in our study” into “we minimized overfitting by considering both biological/clinical and biostatistical importance” in lines 637-638.

  1. LINE 515: “messager” should read “messenger.”

Response: Change was made in line 642.

  1. DISCUSSION: Generally, blood has been shown to have very low levels of DNA hydroxymethylation (see: https://www.ncbi.nlm.nih.gov/pmc/articles/PMC3290782/). Given these low levels compared to other tissues, would the authors expect their detected differential DNA hydroxymethylation to carry over to other tissues, like the heart?

Response: Although we did not have tissue data to address this question, a previous animal experimental study of mouse heart tissue demonstrated hydroxymethylation, particularly in intronic regions, was one epigenetic mechanism underlying the development of dilated cardiomyopathy [8]. To what extent we can extrapolate human findings to animals is unknown. However, Jiang’s research team reported that levels of global DNA 5hmC in the human aortic atherosclerotic tissue were higher in elderly patients with coronary heart disease than controls in 2019 [9] in lines 521-524. Nevertheless, it is intriguing to test our findings in heart and aortic tissues. No changes were made in response to this comment.

References

  1. Szulwach, K.E.; Li, X.; Li, Y.; Song, C.X.; Han, J.W.; Kim, S.; Namburi, S.; Hermetz, K.; Kim, J.J.; Rudd, M.K., et al., Integrating 5-hydroxymethylcytosine into the epigenomic landscape of human embryonic stem cells. PLoS Genet. 2011, 7, e1002154, doi:10.1371/journal.pgen.1002154.
  2. Szulwach, K.E.; Li, X.; Li, Y.; Song, C.X.; Wu, H.; Dai, Q.; Irier, H.; Upadhyay, A.K.; Gearing, M.; Levey, A.I., et al., 5-hmC-mediated epigenetic dynamics during postnatal neurodevelopment and aging. Nat. Neurosci. 2011, 14, 1607-1616, doi:10.1038/nn.2959.
  3. Song, C.X.; Szulwach, K.E.; Fu, Y.; Dai, Q.; Yi, C.; Li, X.; Li, Y.; Chen, C.H.; Zhang, W.; Jian, X., et al., Selective chemical labeling reveals the genome-wide distribution of 5-hydroxymethylcytosine. Nat. Biotechnol. 2011, 29, 68-72 (*corresponding author), doi:10.1038/nbt.1732.
  4. Yu, M.; Hon, G.C.; Szulwach, K.E.; Song, C.X.; Zhang, L.; Kim, A.; Li, X.; Dai, Q.; Shen, Y.; Park, B., et al., Base-resolution analysis of 5-hydroxymethylcytosine in the mammalian genome. Cell 2012, 149, 1368-1380, doi:10.1016/j.cell.2012.04.027.
  5. Bhasin, J.M.; Hu, B.; Ting, A.H., MethylAction: detecting differentially methylated regions that distinguish biological subtypes. Nucleic Acids Res. 2016, 44, 106-116, doi:10.1093/nar/gkv1461.
  6. Glen, S. C-Statistic: Definition, Examples, Weighting and Significance. Availabe online: https://www.statisticshowto.com/c-statistic/ (accessed on July 23).
  7. Wilson, P.W.; D'Agostino, R.B.; Levy, D.; Belanger, A.M.; Silbershatz, H.; Kannel, W.B., Prediction of coronary heart disease using risk factor categories. Circulation 1998, 97, 1837-1847.
  8. Tabish, A.M.; Arif, M.; Song, T.; Elbeck, Z.; Becker, R.C.; Knöll, R.; Sadayappan, S., Association of intronic DNA methylation and hydroxymethylation alterations in the epigenetic etiology of dilated cardiomyopathy. Am. J. Physiol. Heart Circ. Physiol. 2019, 317, H168-h180, doi:10.1152/ajpheart.00758.2018.
  9. Jiang, D.; Sun, M.; You, L.; Lu, K.; Gao, L.; Hu, C.; Wu, S.; Chang, G.; Tao, H.; Zhang, D., DNA methylation and hydroxymethylation are associated with the degree of coronary atherosclerosis in elderly patients with coronary heart disease. Life Sci. 2019, 224, 241-248, doi:10.1016/j.lfs.2019.03.021.